

# 7,8-Dihydroxyflavone protects neurons against oxygen-glucose deprivation induced apoptosis and activates the TrkB/Akt pathway

Qinxiang Zhou[1,2], Hao Tang[2,3], Dingqun Bai[1] and Yuhan Kong[1,2]

[1] Department of Rehabilitation Medicine, The First Affiliated Hospital of Chongqing Medical University, Chongqing, China

[2] Chongqing Key Laboratory of Neurology, The First Affiliated Hospital of Chongqing Medical University, Chongqing, China

[3] Department of Neurology, The Second Affiliated Hospital of Chongqing Medical University, Chongqing, China

## ABSTRACT

**Background**. 7,8-dihydroxyflavone (7,8-DHF), a selective agonist of tropomyosin related kinase receptor B (TrkB), is known to exert protective effects in neurodegenerative diseases. However, the role of 7,8-DHF in TrkB signaling after ischemic stroke has remained elusive.

**Methods**. In the vitro model of ischemic stroke, we investigated the neuroprotective effect of 7,8-DHF through activation of TrkB signaling. Neurons subjected to oxygen and glucose deprivation/reperfusion were treated with the protein kinase inhibitor K252a and a knockdown of TrkB. Cell counting kit-8 (CCK-8) assay, Flow Cytometric Analysis (FACS), TdT-mediated dUTP nick end labeling (TUNEL) assay were conducted for measuring cell viability and numbers of apoptotic cells. And apoptosis-associated proteins were analyzed by Western blotting.

**Results**. Compared with the Control group, OGD/R group revealed lower cell viability by CCK-8 assay FACS and TUNEL assay showed increased rates of neuronal apoptosis. However, 7,8-DHF treatment increased cell viability and reduced neuronal apoptosis. Western blotting indicated upregulated Bax and cleaved caspase-3 and but downregulated Bcl-2 following OGD/R. Whereas 7,8-DHF treatment downregulated Bax and cleaved caspase-3 but upregulated Bcl-2. These changes were accompanied by a significant increase in the phosphorylation of TrkB and Akt following 7,8-DHF administration. However, the administration of K252a and knockdown of TrkB could alleviate those effects.

**Conclusion**. Our study demonstrates that activation of TrkB signaling by 7,8-DHF protects neurons against OGD/R injury via the TrkB/Akt pathway, which provides the evidence for the role of TrkB signaling in OGD-induced neuronal damage and may become a potential therapeutic target for ischemic stroke.

Corresponding authors
Dingqun Bai, baidingqun@163.com
Yuhan Kong,
kongyuhan@hospital.cqmu.edu.cn

## INTRODUCTION

Ischemic stroke has emerged as the leading cause of mortality and long-term disability worldwide (*Virani et al., 2020*). It is pathologically characterized by insufficient blood flow to the brain tissues resulting in oxygen and glucose deprivation (OGD) (*Ryou & Mallet, 2018*). The viability of neurons critically depends on the delivery of oxygen and nutrients *via* blood vessels, a lack of which can result in OGD. Consequently, OGD may result in the failure of both cellular energy machinery and homeostasis and induce several pathophysiological processes, including oxidative stress, neuronal loss, inflammatory responses, and apoptosis (*Tasca, Dal-Cim & Cimarosti, 2015*). Despite substantial efforts being made to explore the pathogenesis, the management of ischemic neuronal injury remains a huge challenge. Although apoptosis is considered to be one of the underlying mechanisms, the precise factors triggering neuronal death and its pathophysiological correlates have remained unclear (*Li et al., 2014*; *Su, Bourdette & Forte, 2012*). Increasing evidence has linked apoptosis to neuronal loss in most neurodegenerative diseases, such as Parkinson's and Alzheimer's diseases (*Waldmeier & Tatton, 2004*). An effective therapeutic approach is required to rescue the neurons from ischemic injury.

BDNF is a neurotrophic molecule that is abundantly expressed in the mammalian central nervous system. It may exert its effects by binding to a transmembrane receptor tyrosine kinase, TrkB, to carry out a variety of biological processes, including neuronal survival (*Numakawa et al., 2010*). Binding of BDNF to TrkB causes dimerization and auto-phosphorylation of specific sites in the receptor, resulting in the activation of downstream signaling. Inhibiting the BDNF/TrkB and TrkB/Akt pathways has been shown in studies to result in OGD-induced neuronal apoptosis (*Du et al., 2020*). By activating Akt, mitochondrial integrity could be preserved. Furthermore, activated Akt could regulate Bcl-2 family members and Bax while also inhibiting pro-apoptotic activities (*Sussman, 2009*). The BDNF/TrkB and TrkB/Akt signaling pathways are critical for neuronal survival. As a result, system It could be a therapeutic strategy to target TrkB/Akt signaling in order to alleviate brain injury caused by an ischemic stroke. However, the poor delivery and short half-life of BDNF *in vivo* limit its clinical applications (*Thoenen & Sendtner, 2002*). 7,8-dihydroxyflavone (7,8-DHF), a small flavonoid, has been identified as an agonist of tropomyosin related kinase receptor (TrkB). 7,8-DHF has emerged as a substitute for BDNF in the past few decades, with better pharmacokinetic properties and a higher TrkB-binding affinity than BDNF (*Andero & Ressler, 2012*). 7,8-DHF can pass the blood–brain barrier and activate TrkB receptors in the brain. 7,8-DHF has been extensively explored for its therapeutic effects in several cell types and disease models (*Jang et al., 2010*). Although studies have reported the ameliorative effects of 7,8-DHF on neurological degenerative disorders, its exact effect and underlying mechanism related to TrkB in ischemic stroke are unclear.

In this study, we investigated whether the activation of TrkB signaling by 7,8-DHF is protective against neuronal glucose and oxygen deprivation. Furtherly we investigated whether 7,8-DHF could promote TrkB/Akt pathway, thereby reducing neuronal injury.

We believe our findings will contribute to alleviating ischemic stroke-induced damage, thereby allowing the development of effective therapeutic strategies.

## MATERIALS & METHODS

### Primary cortical neuron cultures

The cortical neurons were isolated from 1-day-old Sprague–Dawley rats and cultured as described previously (*Roppongi, Champagne-Jorgensen & Siddiqui, 2017*). The rats were obtained from the experimental animal center of Chongqing Medical University (Chongqing, China). The study was approved by Ethics Committee of the First Affiliated Hospital of Chongqing Medical University (Ethical Application Ref: 2020-772). All rats were raised in pair-housed upon arrival in clear, plastic rodent caging at room temperature (26–27 °C) and allowed free access to food and water. Briefly, the rats were decapitated after cleaning with 75% alcohol. The cortical tissues without meninges and blood vessels were carefully extracted from the brain and transferred to an ice-cold buffer composed of 127 mM NaCl, 1.7 mM $NaH_2PO_4$, 5 mM KCl, 2.05 mM $KH_2PO_4$, 10 nM D-glucose, and 100 U mL−1 penicillin/streptomycin (pH 7.4). The tissues were cut mechanically and digested with 0.25% trypsin at 37 °C for 30 min. Next, the tissue sample was neutralized with 10% fetal bovine serum (FBS; Gibco Co., MA, USA) and triturated with a Pasteur pipette. The tissue solution was washed with Hank's balanced salt solution (Gibco), collected on a 400-mesh grid, and centrifuged at 1,000 rpm for 5 min. Afterward, the supernatant was discarded. The cells were seeded in 6-well plates ($1.5 \times 10^6$ cells per well) and 96-well plates ($1 \times 10^4$ cells per well) pre-coated with poly-L-lysine (PLL) in a high-glucose DMEM (Gibco) complete medium containing 10% FBS and $1 \times 10^5$ U/L penicillin and incubated in a humidified chamber with 5% $CO_2$ at 37 °C (Thermo Fisher Scientific Inc., USA). After 5 h, the medium was replaced with a neurobasal medium (Gibco) supplemented with 2% B27 (Gibco) and 1% glutamine (Gibco). Half of the culture medium was replaced every 3 days.

### Oxygen and glucose deprivation/reperfusion

Neurons were subjected to oxygen–glucose deprivation/reperfusion (OGD/R) 7 days after the culture. Briefly, the cells were transferred from the neurobasal-A medium to DMEM without glucose (Gibco) in 6-well plates after being washed thrice by phosphate-buffered saline (PBS). In this medium, the cells were incubated at 37 °C for 4 h in a sealed chamber (Thermo 3111; Thermo Fisher Scientific Inc., USA) in an anaerobic gas mixture filled with 94% $N_2$, 1% $O_2$, and 5% $CO_2$. Cell media were replaced with normal media and cultures for 24 h.

### Drug administration

7,8-DHF (MedChemExpress, USA) and K252a (0.1 μmol) (*Kim & Jin, 2020*) (MedChemExpress, USA) were obtained as a powder and dissolved in dimethyl sulfoxide (DMSO) (one mmol with DMSO concentration adjusted o 0.1%), which were administrated after OGD/R for 0.5 h in an incubator under 95% air/5% $CO_2$ at 37 °C till for 24 h Reperfusion. CCK-8 assay was used to determine a suitable 7,8-DHF concentration

(0.25, 0.5, 0.75, and 1.0 $\mu$M). The cells were seeded into 96-well plates ($2 \times 10^4$ cells/well) and subjected to various treatments as described above. Next, 10 $\mu$L of the CCK-8 solution (Dojindo Molecular Technologies, Inc.) was added to each well, and cells were incubated for 3 h. The absorbance was measured at 450 nm using a spectrophotometer (Thermo Fisher Scientific, Inc.). The cultures were randomly divided into five groups: (1) control group, (2) OGD/R group, (3) OGD/R+7,8-DHF group, (4) OGD/R+7,8-DHF+K252a group, (5) OGD/R+K252a group.

## RNA interference

Small interfering RNA (siRNA) were purchased from Invitrogen (Carlsbad, USA). Neurons at 5 days after the culture were transfected with small interfering RNAs targeting TrkB mRNA (siTrkB)(GUAUCAGCUAUCAAACAAC) or the control-siRNA using Lipofectamine 2000 according to the provider's recommendations. Cells in a humidified atmosphere were randomly divided into Control group, Control +vehicle group and Control +siRNA group. The knockdown efficiency was measured by Western blot analysis after 48 h of culture. Furthermore, the cultures were randomly divided into five groups: (1) control group, (2) OGD/R group, (3) OGD/R+7,8-DHF group, (4) OGD/R+7,8-DHF+siRNA group, (5) OGD/R+7,8-DHF+vehicle group.

## Flow cytometric analysis

Flow cytometry using an Annexin V Apoptosis Detection Kit was used to analyze apoptosis. Briefly, neurons were digested by EDTA-free enzymes and washed with PBS for three times, after being resuspended with 500 $\mu$L binding buffer, neurons were stained with Annexin V FITC apoptosis detection kit (Beyotime). Finally, a FACSCalibur (BD Biosciences, Franklin, NJ, United States) flow cytometer was used to examine the neurons, and the results were analyzed using Cell Quest software (BD Biosciences).

## TUNEL assay

The TUNEL assay was performed to determine the cellular apoptosis rate. The neurons in each group in 6-well plates containing PLL-coated coverslips were washed thrice with PBS and fixed with 4% paraformaldehyde for 10 min at room temperature. The cells were again washed thrice with PBS and twice with Tris-buffered saline (TBS) (8.5 g NaCl, 1.2 g Tris, 0.45 mL acetic acid (98% CH3COOH), and 1 L distilled water) on a shaking table. Next, the neurons were incubated in a mixture of terminal deoxynucleotidyl transferase (TdT) and dUTP (1:9) provided in the TUNEL kit (Boster Biotechnology Wuhan, China) at 37 °C for 2 h. Afterward, the cells were washed thrice with TBS and blocked with 5% bovine serum albumin (BSA) (Sigma Aldrich; Merck KGaA) for 30 min at room temperature. The coverslips were covered with 50 $\mu$L of anti-DIG-biotin (1:100) and streptavidin-biotin complex (SABC) at 37 °C for 30 min. Finally, the neurons were stained with DAPI (Sigma Aldrich) for nuclear staining. fluorescence microscope (Olympus Corporation) was used to observe the number of apoptotic neurons. It was Ratios of numbers of TUNEL-positive nuclei/ total number of nuclei that was calculated as Percentages of TUNEL-positive cells.

## Western blotting

To extract the total protein, cells were washed thrice with PBS and then homogenized in a lysis buffer (Beyotime Institute of Biotechnology) containing protease and phosphatase inhibitor cocktails (Roche, Germany). The cell lysate was set aside for 30 min on ice, after which it was centrifuged at 16,000 rpm for 30 min at 4 °C. The supernatant was used to determine the protein concentration using the BCA Protein Assay Kit (Beyotime Institute of Biotechnology) and 1 μg/μL sample was run on 10% and 12% SDS-PAGE, following which the proteins were transferred to nitrocellulose membranes and blocked with 7% milk in TBS and incubated overnight at 4 °C with the following primary antibodies: anti-pTrkB (1:1,000; Cell Signaling Technology), anti-TrkB (1:1,000, Cell Signaling Technology), anti-pAkt (1:1,000; Cell Signaling Technology), anti-Akt (1:1,000; Cell Signaling),GAPDH (1:1,000; Proteintech China), anti-Bcl-2 (1:1,000; Abcam), anti-cleaved-caspase 3 (1:1,000; Abcam), and anti-Bax (1:1,000; Abcam) overnight at 4 °C and washed three times in TBST (Tris-buffered saline and Tween). The protein bands were incubated with secondary antibodies (1:5,000; anti-rabbit IgG antibodies, Abcam) for 1 h at room temperature. After three washes as indicated above, the protein signal was detected using a chemiluminescence kit (Perkin-Elmer). The relative protein levels were quantified using the ImageJ software (NIH, Bethesda, MD, USA).

## Statistical analysis

Data were obtained from at least three independent experiments with similar results and are expressed as mean ± standard deviation (SD). The results were analyzed using GraphPad Prism 6.0 (GraphPad, San Diego, CA, USA). One-way ANOVA with a Tukey's multiple comparisons post hoc test was used to assess differences among the groups. A value of $p < 0.05$ was considered significant.

# RESULTS

## Determination of a suitable concentration of 7,8-DHF and effects on neuronal apoptosis

The effect of different concentrations of 7,8-DHF on the neurons after OGD/R was assessed in terms of cell viability using the CCK-8 assay. The appropriate concentration that resulted in higher cell viability, when compared with the OGD/R group, was determined (Fig. 1A). Compared with the control group, the cell viability rates significantly decreased in the OGD/R group ($p < 0.05$). Further, the groups treated with four different concentrations of 7,8-DHF (0.25, 0.5, 0.75, and 1 μmol) had higher cell viability rates than the OGD/R group ($p < 0.05$). The 0.5 μmol 7,8-DHF group had a better effect among the four concentration groups. Therefore, it was selected for our further experiments and to explore the effects of 7,8-DHF on neurons under OGD/R. We evaluated neuronal apoptosis in three groups (control, OGD/R, OGD/R+7,8- DHF) to observe the effects of 7,8-DHF administration on neurons (Figs. 1B–1C). The results of Flow Cytometric Analysis showed that OGD/R induced significant neuronal apoptosis. However, the administration of 7,8-DHF led to dramatically less neuronal apoptosis ($p < 0.001$).

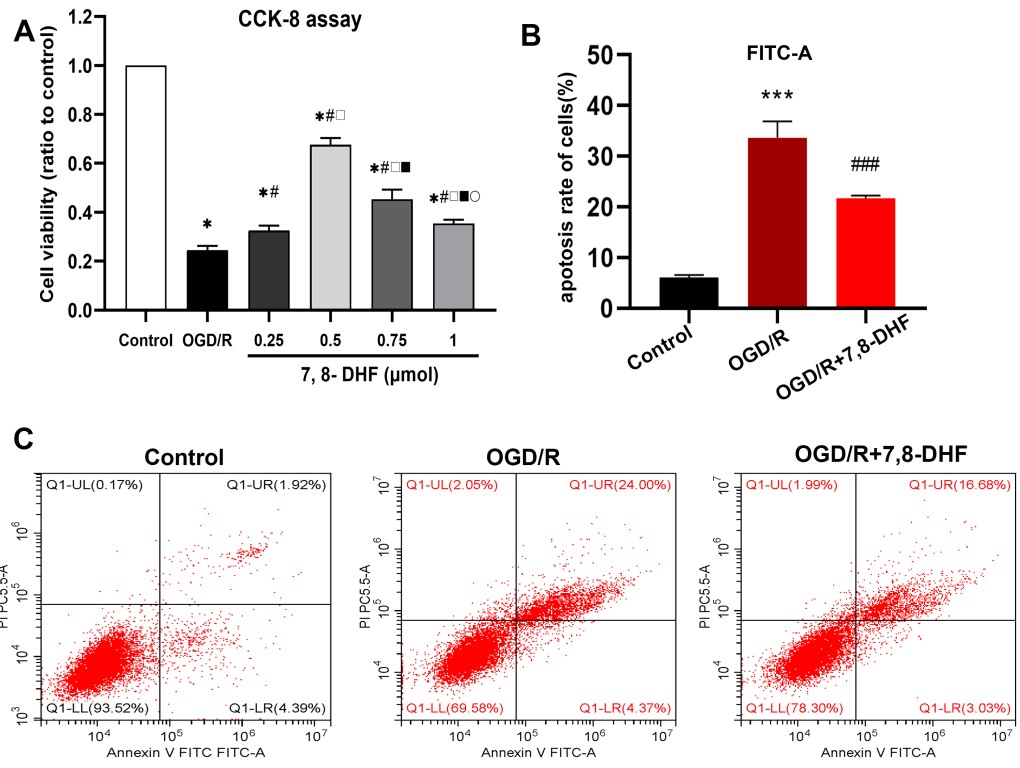

**Figure 1** **Determination of a suitable concentration of 7,8-DHF and Flow Cytometric Analysis.** (A) The cell viability was assessed using a cell counting kit-8 (CCK-8) assay compared with the control group. The cell viability was significantly increased in the OGD/R+7,8-DHF groups as compared with that in the OGD/R group and peaked in the 0.5 μmol OGD/R+7,8-DHF group. *$p < 0.05$. *vs* the control group; # $p < 0.05$ *vs* OGD/R group; □ $p < 0.05$ *vs* 0.25 μmol 7,8-DHF+OGD/R group; ■ $p < 0.05$ *vs* 0.75 μmol OGD/R+7,8-DHF group; ○ $p < 0.05$ *vs* 1 μmol OGD/R+7,8-DHF group. (B–C) Neuronal apoptosis is analyzed by Flow Cytometric Analysis. Mean ± standard deviation (SD) values from 24 independent experiments are presented. ***$p < 0.001$. *vs* The Control group, ###$p < 0.001$. *vs* OGD/R group. Differences between the groups were analyzed using one-way ANOVA followed by Tukey's post hoc test.

## The effects of 7,8-DHF on inhibiting OGD/R-induced cell apoptosis and regulating apoptosis-related protein expression

As shown in Figs. 2A and 2C, the percentage of TUNEL-positive cells in the OGD/R group were significantly increased compared with that in the control group ($p < 0.001$), indicating the markedly increased apoptosis rate. Notably, fewer apoptotic cells were observed in the OGD/R+7,8-DHF group than in the OGD/R group ($p < 0.01$). These results demonstrated that 7,8-DHF significantly inhibited cell apoptosis. Furthermore, three important indicators of the apoptotic pathway were investigated. First, we examined the expression of Bcl-2, an anti-apoptotic protein, and Bax, a pro-apoptotic protein, and cleaved-caspase 3 (Fig. 2B). The OGD/R group showed a significantly decreased level of Bcl-2 and increased levels of Bax and cleaved-caspase 3 when compared with the control group ($p < 0.05$). However, 7,8-DHF treatment elevated the Bcl-2 expression and reduced the Bax expression. In addition, 7,8-DHF treatment inhibited OGD/R-induced increased expression of cleaved-caspase 3 ($p < 0.05$).

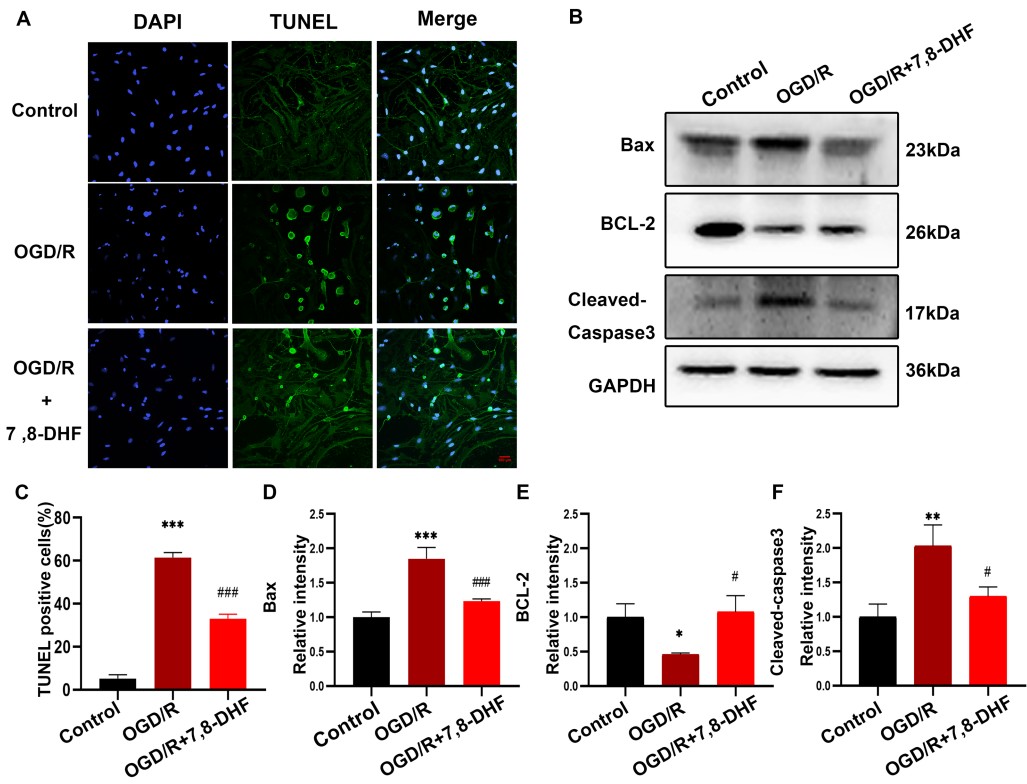

**Figure 2** **7,8-DHF inhibits OGD/R-induced cell apoptosis.** (A) The TUNEL staining of neurons in the control, OGD/R, and OGD/R+7,8-DHF groups. (C) The semiquantitative analysis of TUNEL positive neurons in the three groups. (B) Western blotting of B-cell lymphoma 2 (Bcl-2), Cleaved-Caspase-3, and Bcl-2-associated X (Bax) expression in control, OGD/R, and OGD/R+7,8-DHF groups. The semiquantitative analysis of Bax (D), Bcl-2 (E), and cleaved-caspase 3 (F). Scale bar = 50 μm. Mean ± standard deviation (SD) values are presented. *$p < 0.05$, **$p < 0.01$, ***$p < 0.001$. *vs* the control group.; #$p < 0.05$, ##$p < 0.01$, ###$p < 0.001$. *vs* OGD/R group. Differences between the groups were analyzed using one-way ANOVA followed by Tukey's post hoc test.

## Activities of TrkB and Akt induced by 7,8-DHF after OGD/R

To analyze the effect of 7,8-DHF on TrkB and its associated downstream signaling, pharmacologically inhibiting TrkB by using K252a was conducted. We examined whether 7,8-DHA treatment could promote the expression of TrkB and Akt with the treatment of K252a. As shown in Figs. 3A and 3B, expressions of pTrkB and pAkt were reduced after OGD/R ($p < 0.05$). 7,8-DHF treatment increased relative intensity of pTrkB/TrkB and pAkt/Akt expression. However, when compared with OGD/R+7,8-DHFgroup, 7,8-DHF group relative intensity of pTrkB/TrkB and pAkt/Akt levels were decreased in OGD/R+7,8-DHF+K252a group, which indicated K252a partially blocked the progress induced by 7,8-DHF. This also affects the expression of apoptosis-related proteins (Fig. 4).

## The role of TrkB in TrkB/Akt signaling activated by 7,8-DHF

Furtherly, to confirm the necessities of TrkB for 7,8-DHF and the related TrkB is within neurons. We utilized a small interfering RNA targeting TrkB mRNA (siTrkB) to knock down TrkB in neurons. siTrkB neurons had relatively reduction in TrkB expression (Fig. 5A). We

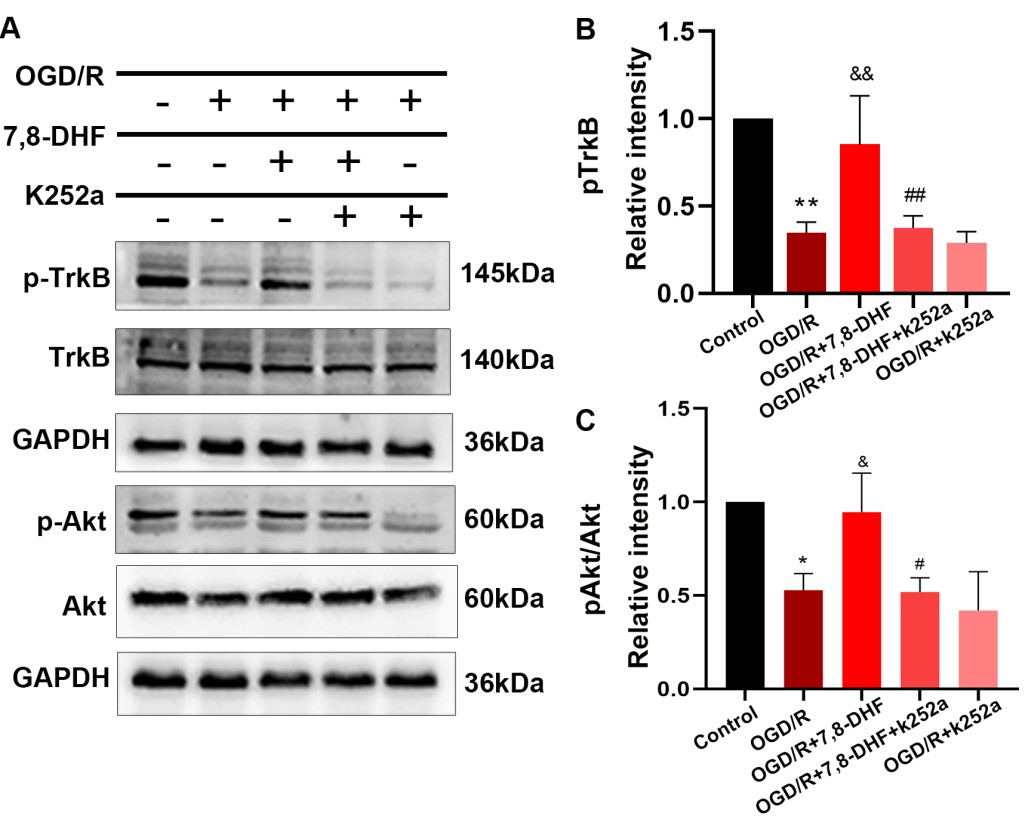

**Figure 3** **Effect of 7,8-DHF on expression of TrkB and Akt.** (A) Western blotting of TrkB, pTrkB, Akt and pAkt in five groups. The semiquantitative analysis of pTrkB (B) and pAkt/Akt (C). $*p < 0.05$, $**p < 0.01$. *vs* the control group; Mean $\pm$ standard deviation (SD) values are presented. $\& p < 0.05$, $\&\& p < 0.01$. *vs* OGD/R group; $\#p < 0.05$, $\#\#p < 0.01$. *vs* OGD/R+7,8-DHF group. Differences between the groups were analyzed using one-way ANOVA followed by Tukey's post hoc test.

examined whether 7,8-DHA treatment could still activate TrkB/Akt signaling. As shown in Fig. 5C, when compared with OGD/R+7,8-DHFgroup, 7,8-DHF group, TrkB, pTrkB and pAkt/Akt levels were decreased in OGD/R+7,8-DHF+siTrkB group ($p < 0.001$), which indicated the effects of 7,8-DHF treatment were partially abolished by a knockdown of TrkB protein expression.

## Protective effects of 7,8-DHF by activating TrkB/Akt signaling

To better understand the protective mechanism of 7,8-DHF against OGD/R induced apoptosis, K252a treatment (Fig. 4) and a knockdown of TrkB (Fig. 5) was performed to confirm the effect of 7,8-DHF in preventing OGD/R-induced apoptosis in neurons, and we found that OGD/R significantly induced neural apoptosis. However, relative to that of the OGD/R group, 7,8-DHF treatment effectively attenuated OGD/R-induced apoptosis. Nevertheless, the apoptosis inhibitory effect of 7,8-DHF in neurons can be attenuated by K252a and a knockdown of TrkB. As shown Figs. 4 and 5, lower Bax, Cleaved-caspase-3, and higher BCL-2 expressions were observed in the 7,8-DHF treatment groups than those in OGD/R group ($p < 0.05$). However, those effects were suppressed by treatment

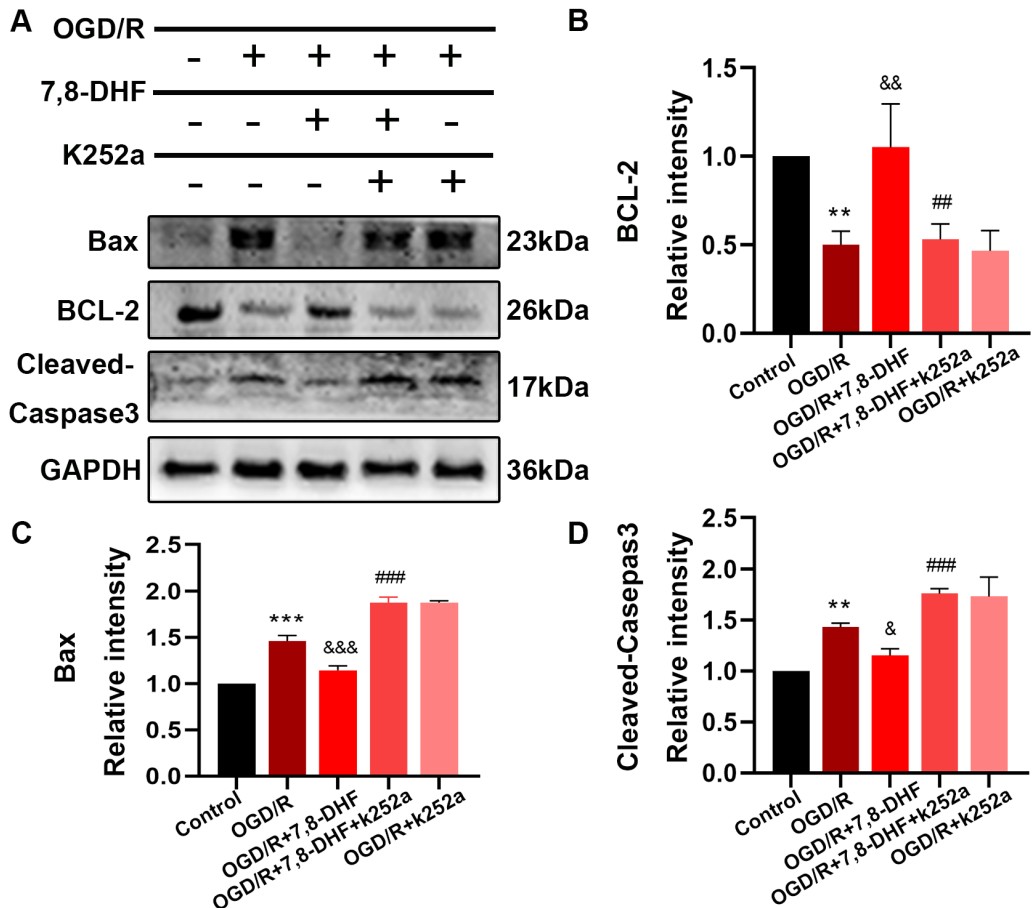

**Figure 4** **Effect of 7, 8-DHF and K252a on expression of apoptosis related proteins.** (A) Western blotting showing the expression of Bax, BCL-2 and Cleaved-Caspase3 in five groups. The semiquantitative analysis of Bax (C), Bcl-2 (B), and cleaved-caspase 3 (D). Mean ± standard deviation (SD) values are presented. $**p < 0.01$, $***p < 0.001$. *vs* the control group; $\&p < 0.05$, $\&\&p < 0.01$, $\&\&\&p < 0.001$. *vs* OGD/R group; $\#\#p < 0.01$, $\#\#\#p < 0.001$. *vs* OGD/R+7,8-DHF group. Differences between the groups were analyzed using one-way ANOVA followed by Tukey's post hoc test.

with K252a and a knockdown of TrkB (Figs. 5G–5I). In contrast, Bax, Cleaved-caspase3 levels were increased and BCL-2 levels were decreased in OGD/R+7,8-DHF+K252a group and OGD/ R+7,8-DHF+siTrkB group when compared with OGD/R+ 7,8-DHF group ($p < 0.01$). As a result, 7,8-DHF was effective in protecting neurons from preventing OGD-induced apoptosis and TrkB/Akt signaling was necessary for the neuroprotective functions of 7,8-DHF.

## DISCUSSION

### Anti-apoptotic effects of 7,8-DHF in cerebral ischemia

Cerebral ischemic penumbra (CIP) refers to the injured ischemic brain tissue and is a crucial target of therapeutics (*Jackman & Iadecola, 2015*). An ischemic penumbra is characterized by the activation of apoptotic pathways and inflammatory responses resulting in neuronal

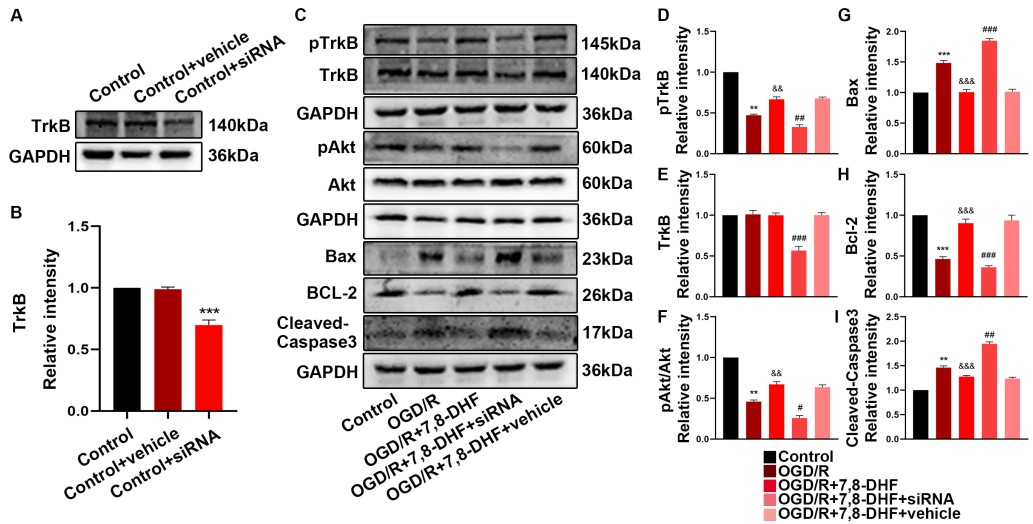

**Figure 5** **The role of TrkB in neurons after 7,8-DHF treatment.** (A) Western blotting showing a knockdown of TrkB. (B) The semiquantitative analysis of TrkB. (C) Effect of 7,8-DHF on related proteins with a knockdown of TrkB. The semiquantitative analysis of pTrkB (D), TrkB (E), pAkt/Akt (F), Bax (G), Bcl-2 (H) Cleavd-Caspase3 (I). Mean ± standard deviation (SD) values are presented. **$p < 0.01$, ***$p < 0.001$. vs the control group; &&$p < 0.01$, &&&$p < 0.001$. vs OGD/R group; #$p < 0.05$, ##$p < 0.01$, ###$p < 0.001$. vs OGD/R+7,8-DHF group. Differences between the groups were analyzed using one-way ANOVA followed by Tukey's post hoc test.

apoptosis (*Broughton, Reutens & Sobey, 2009*; *Kawabori & Yenari, 2015*). The damaged neurons in this region can be reversed to a healthy state if treated appropriately. The therapeutic strategy to rescue the ischemic penumbra primarily focuses on inhibiting apoptosis and modulating inflammatory responses (*Broughton, Reutens & Sobey, 2009*; *Macrez et al., 2011*). OGD/R results in ischemic neuronal damage by inducing apoptosis, neuroinflammatory responses, and autophagy (*Shan et al., 2019*). In the present study, Flow Cytometric Analysis and the TUNEL assay revealed that OGD/R induced excessive apoptosis. However, the treatment with 7,8-DHF reduced numbers of apoptotic cells. Members of the B cell lymphoma-2 (Bcl-2) family, such as Bcl-2, Bax, and cleaved-caspase 3 play a vital role in regulating apoptosis, whereas their abnormal expression can promote apoptosis (*Czabotar et al., 2014*; *Liu et al., 2018*). 7,8-DHF treatment groups showed upregulated Bcl-2, and downregulated Bax and cleaved-caspase 3, implying that it protected the neurons against apoptosis. These findings indicated that 7,8-DHF, with strong neuroprotective functions, markedly inhibited OGD-induced injury by suppressing apoptosis.

## Roles of TrkB/Akt signal pathway and 7,8-DHF in protecting neurons from ischemic injury in central nervous system

BDNF belongs to the family of neurotrophies and is widely distributed in the peripheral and central nervous systems. It binds to a specific receptor, TrkB, and this complex exerts its neurotrophic effects by providing nutritional support to the neurons (*Ji et al., 2005*). The binding of BDNF to TrkB induces dimerization and auto-phosphorylation of

specific sites in the receptor, consequently initiating Akt pathway and PLC pathways to regulate intracellular Ca2+ and inhibit neuronal cell apoptosis. Therefore, the BDNF/TrkB signaling and downstream TrkB/Akt signaling protects the injured neurons and promotes the recovery of cerebral ischemia (*Eberhardt et al., 2006*; *Mantilla et al., 2013*; *Yoshii & Constantine-Paton, 2010*). Deficiencies in the BDNF/TrkB/Akt activity have been identified in several neurodegenerative diseases such as Alzheimer's disease (*Hu & Russek, 2008*). 7,8-DHF is considered to functionally mimic BDNF, which could activate BDNF/TrkB pathway and the downstream pathway (*Jang et al., 2010*). We found that OGD/R-triggered reduction in the pathway was reversed after 7,8-DHF treatment. In addition, the reduced number of apoptotic cells and the downregulated levels of apoptotic proteins in our study following the treatment of neurons with 7,8-DHF indicated that the OGD/R-induced neuronal apoptosis was alleviated. In the present study, we observed downregulated activity of the TrkB/Akt pathway in the neurons after OGD/R, similarly to the results following TrkB/Akt deficiencies observed in other CNS diseases. These findings suggested that TrkB/Akt signaling could be one of the mechanisms underlying OGD/R-induced injuries. we asked whether downstream activation of TrkB receptors by 7,8-DHF was required for neuroprotection. So, we took a pharmacological approach using TrkB receptor inhibitor K252a and knocked down TrkB to inhibit TrkB/Akt signaling pathway. Our study found that 7,8-DHF could attenuate OGD/R-related phenomena, including enhanced apoptosis and downregulation of TrkB/Akt signaling, whose effects were partially alleviated by K252a and a loss of TrkB protein. The results suggested that TrkB/Akt pathway is necessary for protective role of 7,8-DHF. Thus, 7,8-DHF activated the TrkB/Akt signaling pathway and protected the injured neurons. These findings indicated that 7,8-DHF protects neurons against ischemic injury and promotes neuronal survival by exerting physiological effects similar to those of BDNF (*Jang et al., 2010*).

## Potential Role of 7,8-DHF in neurological disorders

7,8-DHF is a natural flavone derivative that is obtained from Godmania aesculifolia, Tridax procumbens, and certain other plants. Several of its biological effects have been investigated, with increasing attention being given to its effects on neurodegeneration and neuroprotection (*Obianyo & Ye, 2013*; *Ragen et al., 2015*; *Zeng et al., 2013*). Certain studies have reported the therapeutic effects of 7,8-DHF on diseases related to CNS, such as Alzheimer's disease (*Chen et al., 2018*), Parkinson's disease (*Sconce et al., 2015*), cognitive dysfunction (*García-Díaz Barriga et al., 2017*), and depression (*Amin et al., 2020*). However, the literature on its role in stroke is little. Therefore, more studies need to be conducted to prove its beneficial effects. Neuroprotective and anti-apoptotic effects of 7,8-DHF have been described in cellular and animal models deficient for TrkB expression (*Choi et al., 2013*; *Han et al., 2014*; *Ryu et al., 2014*). However, the molecular mechanism of 7,8-DHF-induced TrkB activity varies in different diseases (*Jiang et al., 2013*; *Todd et al., 2014*). In this study, 7,8-DHF upregulated pTrkB and the downstream pAkt in the neurons after OGD/R and protected neurons from apoptosis. Furthermore, we found TrkB/Akt pathway signaling is required for the possible underlying mechanism of protective roles of 7,8-DHF in ischemic stroke. These findings suggest that the intact TrkB/Akt signaling

cascade is required for 7,8-DHF to have neuroprotective effects. This also suggests that the positive feedback of the BDNF/TrkB/Akt signal is important for neuronal survival (*Kowiański et al., 2018*). As a result, the BDNF/TrkB/Akt signaling axis is a promising target for neuroprotective research and drug discovery. There were applications of recombinant BDNF in clinic, however, the effect is not satisfactory (*Ochs et al., 2000*), probably due to its limited delivery, short half-life, and inability to cross the blood–brain barrier. Alternatively, 7,8-DHF could pass through the blood–brain barrier and can be administered *via* various routes such as oral, intraperitoneal(i.p.) and intramuscular (i.m.), it is unlikely to elicit an immune response and mimics the biological functions of BDNF by binding directly to the TrkB receptor (*Andero et al., 2011*). As a result, 7,8-DHF is expected to be a therapeutic target with broader clinical applications than BDNF.

## CONCLUSIONS

In conclusion, our study demonstrated the protective effects of 7,8-DHF against neuronal apoptosis after OGD/R injury. The results showed that these effects were mediated by the interaction of 7,8-DHF with the TrkB/Akt signaling pathway. We believe that 7,8-DHF could serve as a potential therapeutic target for clinical applications in ischemic stroke.

## ACKNOWLEDGEMENTS

We appreciate the Chongqing Key Laboratory of Neurology, Chongqing, China for providing the study with the related experimental equipment.

### Funding

This study was supported by the National Natural Science Foundation of China (No. 81401865) and the Chongqing Municipal Science and Technology commission (No. cstc 2019jcyj-msxmX0339). The funders had no role in study design, data collection and analysis, decision to publish, or preparation of the manuscript.

### Grant Disclosures

The following grant information was disclosed by the authors:
National Natural Science Foundation of China: 81401865.
The Chongqing Municipal Science and Technology commission: cstc 2019jcyj-msxmX0339.

### Competing Interests

The authors declare there are no competing interests.

### Author Contributions

- Qinxiang Zhou conceived and designed the experiments, performed the experiments, analyzed the data, prepared figures and/or tables, authored or reviewed drafts of the paper, and approved the final draft.

- Hao Tang performed the experiments, analyzed the data, prepared figures and/or tables, and approved the final draft.
- Dingqun Bai conceived and designed the experiments, authored or reviewed drafts of the paper, and approved the final draft.
- Yuhan Kong conceived and designed the experiments, analyzed the data, prepared figures and/or tables, authored or reviewed drafts of the paper, and approved the final draft.

### Animal Ethics

The following information was supplied relating to ethical approvals (i.e., approving body and any reference numbers):

The Ethics Committee of The First Affiliated Hospital of Chongqing Medical University provided full approval for this research.

### Data Availability

The raw data is available in the Supplemental Files.

### Supplemental Information

Supplemental information for this article can be found online at http://dx.doi.org/10.7717/peerj.12886#supplemental-information.

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
