# Peer review of ",8-Dihydroxyflavone protects neurons against oxygen-glucose deprivation induced apoptosis and activates the TrkB/Akt pathway"

_PeerJ, doi:10.7717/peerj.12886_

## Round 0.1 · original submission · Major Revisions

Please consider carefully the Reviewers' comments.

Reviewer 1 ·

Basic reporting

In the current study, the authors evaluated the protective effect of 7,8-DHF on primary neurons triggered by OGD. The found that 7,8-DHF effectively protects neurons by activating BDNF/p-TrkB pathway , associated with anti-apoptotic Bcl2 escalation and pro-apoptotic BAX repression. Overall, this study is interesting and fits with previous observations. However, the study is too preliminary. Several key issues are needed to be clarified before acceptance for publication.

Experimental design

Major concerns:
1) in Fig 1, the authors need to show 7,8-DHF displays agonistic effect on p-TrkB/TrkB; p-Akt/Akt; p-MAPK/MAPK in primary neurons in the presence of OGD and absence of OGD.
2) in Fig2-4, the authors need to include Trk receptor inhibitor K252a, to see whether 7,8-DHF indeed to exert this protective role via TrkB receptor or other unknown pathway or effectors.
3) In Fig 2-4, the authors need to knock down TrkB receptor in primary neurons to see whether depletion of TrkB abolishes 7,8-DHF beneficial roles.

Minor concerns:
1) what p-TrkB antibody was employed? usually, a couple of different p-TrkB antibodies against p-706 or p-816 are necessary for demonstrating TrkB activation.
2) the inflammatory cytokines are need to be determined as the authors indicated BDNF exerts anti-inflammatory effect.

Validity of the findings

The quality fo p-TrkB needs to improve!

BDNF in the cell lysate or secreted in the media should be validated by ELISA kits

Additional comments

7,8-DHF was not explored for a few decades. It was reported in 2010. It is merely 11 years. The description in the text is not accurate!

Reviewer 2 ·

Basic reporting

In this study, the authors aim to investigate the possible effects of 7,8-DHF treatment on neurons after OGD/R and explore the potential mechanisms. They concluded that 7,8-DHF protects neuron from OGD damage via activating BDNF-TrkB signaling pathway. In general, this study is well organized. The methods are easy to follow. However, this study can be improved on the following aspects.
1. Moderate improvement in grammar will be needed in this manuscript. And I see some dashes within words. For instance, line 239, “cor-tical” and line 254, “expo-sure”.
2. I don’t think the Hoechst staining method that the authors used can differentiate apoptotic cells. The authors indicate that they fixed the cells first and used Hoechst dye. 33342 Hoechst is a nuclear-specific dye that stains every nuclear it meets, just as DAPI. I see the authors cited one paper (Wan et al). However, even in the cited paper, Wan et al did not use Hoechst for apoptosis assay. They used Annexin/PI staining for cell apoptosis, and it was detected by FACS. I would strongly suggest the authors to do so. In addition, I did not see the described fragmented nuclei, condensed chromatin, etc, neither. The resolution and magnification of images shown are not sufficient to provide these details.
3. Another issue for this study is that the data provided is not sufficient to show 7,8-DHF rescues cells via BDNF-TrkB signaling pathway. I would suggest the authors to knockdown or knockout TrkB and see if 7,8-DHF is still be able to decrease cell apoptosis after OGD/R.
4. Furthermore, I see 7,8-DHF increases BDNF after OGD/R. Does 7,8-DHF also increase BDNF without OGD/R? From the western blot results in Figure 4, it is hard to tell whether enhanced TrkB phosphorylation is a result of 7,8-DHF directly or it is because 7,8-DHF increased BDNF, and that increase of BDNF enhanced TrkB phosphorylation. In addition, total TrkB blot is needed to confirm whether TrkB signaling is elevated after 7,8-DHF supplementation.

Experimental design

Please see details in Basic reporting.

Validity of the findings

Please see details in Basic reporting.

---

## Round 0.2 · Major Revisions

There are still major concerns to consider.

Reviewer 2 ·

Basic reporting

In this version, the authors have addressed most part of my comments. They did better proof reading, reverified their apoptosis assay by FACS. Following are the comments:
1. I can still find typos. For instance, it is post-hoc test, not post-test.
2. I can understand that generating knockout cell line is time consuming. However, knockdown experiment won’t take long, and it is effective to show your target of interest has a role in the proposed mechanism directly. I would strongly suggest the authors to knockdown TrkB and see if 7,8-DHF still decreases apoptosis. The authors can leave the K-252a data in this study. After all, it would be better when the authors prove TrK is important in their study by both genomic and pharmaceutical approach. But keep in mind, that K-252a also inhibits PKC, PKA, Ca2+/calmodulin-dependent kinase type II, and phosphorylase kinase, respectively, with different IC50. When it inhibits TrK, the activity of other protein kinases will also be affected. Especially, the authors used K-252a at a high concentration (100 nM). That is more than 30 times higher than its IC50 for TrK. Is it a safe dose for cell?
3. In addition, the authors should also specify the statistical methods used in their figure legends. They only put one-way ANOVA in their figure legends. However, according to the Statistical analysis part, I learned they also used either Bonferroni or Dunnett’s test for multiple comparison. In this case, please add the information of post-hoc test in and specify which specific post-hoc test is used for each figure.

Experimental design

Additional experiments are needed.

Validity of the findings

Most of the findings are scientifically sound.

---

## Round 0.3 · accepted · Accept

The authors addressed properly the reviewer's concerns.

Reviewer 2 ·

Basic reporting

The authors addressed my concerns from last review. I have one suggestion for the authors. Neuron is usually picky on transfection reagents and methods. Lipofectamine 2000 is not the best option for neuron transfection. This has been reflected on their results in Figure 5. The siRNA knockdown efficiency is low. To optimize transfection efficiency, electroporation is recommended. In the future, I would suggest the authors to read more published papers and have a better knowledge of the cell they are going use, before they start experiments. If it is applicable, this manuscript can be considered for publication.

Experimental design

No comment

Validity of the findings

No comment